# TLR4 Asp299Gly SNP (rs4986790) Protects from Periodontal Inflammatory Destruction by Altering TLR4 Susceptibility to LPS Stimulation

**DOI:** 10.3390/biology14070894

**Published:** 2025-07-21

**Authors:** Franco Cavalla, Claudia C. Biguetti, Ariadne Letra, Renato M. Silva, Alexandre R. Vieira, Franz J. Strauss, Gustavo P. Garlet

**Affiliations:** 1School of Dentistry, University Andres Bello, Santiago 7590924, Chile; 2Conservative Dentistry Department, University of Chile, Santiago 8820808, Chile; 3Bauru School of Dentistry, University of Sao Paulo, Bauru 14040-902, Brazil; claudia.biguetti@utrgv.edu (C.C.B.); garletgp@usp.br (G.P.G.); 4Research Lab of Regenerative Medicine, The University of Texas Rio Grande Valley, Edinburg, TX 78539, USA; 5Departments of Oral and Craniofacial Sciences, University of Pittsburgh School of Dental Medicine, Pittsburgh, PA 15260, USA; ariadneletra@pitt.edu (A.L.); vieiraa23@ecu.edu (A.R.V.); 6School of Dental Medicine, East Carolina University, Greenville, NC 27858, USA; 7Clinic of Reconstructive Dentistry, Center of Dental Medicine, University of Zurich, 8006 Zurich, Switzerland; franz.strauss@zzm.uzh.ch; 8Health Sciences School, Universidad Autonoma de Chile, Santiago 7500912, Chile

**Keywords:** periodontitis, TLR4, SNP, rs4986790, inflammation, LPS, IL-8, immune response

## Abstract

Periodontitis is a common disease that affects the gums and the bone supporting the teeth, often leading to tooth loss. It results from a complex interaction between bacteria and the body’s immune response. Some people are more susceptible than others, and genetics may play a role. In this study, we focused on a genetic variation (called a polymorphism) in a gene known as TLR4, which helps the immune system recognize harmful bacteria. We examined over 1400 individuals from different populations and found that people carrying this specific variation, known as Asp299Gly, were less likely to develop severe periodontal damage. To understand why, we studied cells that had been modified to carry either the normal or the variant version of the gene in the lab. We found that the variant version responded differently to bacterial signals, which may help explain its protective effect. These results suggest that genetic factors can influence how the body reacts to bacteria in the mouth, potentially offering natural protection against periodontitis. This information could one day help identify people at a lower or higher risk for gum disease and lead to more personalized approaches to prevention and treatment.

## 1. Introduction

Periodontitis is a multifactorial non-communicable disease associated with dysbiotic biofilm and characterized by the progressive destruction of the tooth-supporting apparatus. There are two potential points of contact between the dysbiotic biofilm and periodontal tissues: the gingival sulcus and the root canal system [1]. In both cases, infectious agents bypass physical barriers and gain access to the underlying connective tissues, triggering a robust immune response. This response is responsible for the metabolic alterations in the tissues that lead to the inflammatory destruction of the periodontal ligament, radicular cementum, and alveolar bone [2].

Several environmental and intrinsic host factors can modulate the nature of the immune response and, in turn, influence the course of the disease. The host’s ability to respond effectively to the bacterial challenge—organizing a defense that limits microbial spread while avoiding extensive tissue damage—is the key mechanism underlying the pathogenesis of inflammatory periodontal destruction [3]. A competent and regulated immune response should, in theory, control microbial proliferation while preserving the metabolism, function, and structural integrity of the periodontal tissues. However, during the progression of inflammatory periodontal destruction, the persistent presence of microbes and their products leads to a self-amplifying loop of immune and inflammatory responses, ultimately resulting in tissue breakdown [4].

Extensive evidence supports the strong involvement of genetic factors in susceptibility to periodontal tissue destruction. Nevertheless, despite considerable efforts, the identification of specific genetic risk factors has remained elusive [5].

TLR4 is a multiprotein complex that serves as the primary innate immune receptor for lipopolysaccharides [6]. The TLR4 Asp299Gly SNP (rs4986790) is a non-synonymous polymorphism implicated in altered infection responses across various diseases; however, its role in periodontitis remains controversial [7,8,9].

The objective of this study was to evaluate the association between the TLR4 Asp299Gly single-nucleotide polymorphism (rs4986790) and susceptibility to periodontal inflammatory destruction. We also aimed to explore the functional consequences of this variant in vitro. We hypothesized that the Asp299Gly polymorphism alters TLR4-mediated immune responses to bacterial LPS, resulting in a modified inflammatory profile that may confer protection against periodontal tissue breakdown.

Here, we report a genetic association study evaluating the TLR4 Asp299Gly SNP (rs4986790) and its relationship to susceptibility to periodontal inflammatory destruction in an exploratory Brazilian cohort of periodontitis patients and controls (*n* = 570), as well as in three independent validation populations totaling 1410 individuals (528 cases and 882 controls). This SNP was selected due to its previously suggested role in modulating host immune responses to bacterial LPS in periodontal disease [9]. In addition, we performed in vitro functional experiments to explore potential mechanisms by which this polymorphic receptor might influence periodontal destruction.

## 2. Materials and Methods

### 2.1. Participants

The exploration sample was recruited in the São Paulo state, southeastern region of Brazil, from patients scheduled for treatment at the Dentistry School University of Ribeirão Preto (RBP). Patients were examined by an experienced periodontist and scored for bleeding on probing (BOP), probing depth (PD), and clinical attachment loss (CAL). Enrolled subjects provided informed consent that was approved by the institutional review board. Subjects were excluded from the study if they presented a tobacco smoking habit (including former smokers), medical history indicating evidence of known systemic modifiers of periodontal disease, and/or had received periodontal therapy in the previous 2 years. No strategy was used to identify subpopulations (population stratification) or population relatedness among the recruited subjects. In total, 570 subjects were recruited. After the diagnostic phase, subjects were categorized into healthy (H; *n* = 384) or chronic periodontitis patients (CP; *n* = 186) [10,11].

Validation samples were recruited in Pittsburgh (PA, USA) (PIT) [*n* = 274; 70 cases, 204 controls], Houston (TX, USA) (HOU) [*n* = 289; 158 cases, 131 controls], and Guarulhos/Bauru (SP, Brazil) (SAO) [*n* = 277; 114 cases, 163 controls]. The total sample size, including the original sample and the three replication samples, was *n* = 1410; 528 cases and 882 controls.

The Pittsburgh (PA, USA) sample was obtained from the Dental Registry and DNA Repository of the School of Dental Medicine University of Pittsburgh screening for subjects with BOP > 1.5, PD > 3 mm and CAL > 3 mm for the chronic periodontitis sample (*n* = 70) and age/gender-matched healthy controls (*n* = 204). These samples were from consented individuals from Pittsburgh and neighboring areas (average age 42.9 years). Individuals on the registry reflected the demographic distribution of the City of Pittsburgh (70% White, 20% African American, and 10% other ethnic groups). Approximately 60% of the subjects were females. Most of the individuals were from a low socioeconomic stratum, based on their insurability [12].

The Houston (TX, USA) sample comprised consented individuals with apical periodontitis (*n* = 158), presenting with periapical rarefaction, characterized radiographically by the disappearance of the periodontal ligament space and discontinuity of the lamina dura.

Healthy controls were recruited at the surgery clinic from individuals referred for extraction of premolars due to orthodontic reasons (*n* = 131). Control subjects were clinically examined for gingival inflammation (BOP < 1), had PD < 2 mm, and presented no radiographic signs of marginal bone loss or periapical rarefaction.

The Guarulhos/Bauru (SP, Brazil) sample consisted of consented individuals with previously untreated periodontitis referred to the periodontal clinic of Guarulhos University (*n* = 114). All subjects were in good general health and presented with at least 15 teeth, excluding third molars and teeth indicated for extraction. All subjects were diagnosed with generalized chronic periodontitis based on the current classification of the American Academy of Periodontology [13]. The inclusion criteria were as follows: ≥30 years of age and a minimum of six teeth with at least one site each with PD and CAL ≥ 5 mm, as well as at least 30% of the sites with PD and CAL ≥ 4 mm and BOP [14]. Controls were age/gender-matched subjects recruited among the professors and graduate students of the University of São Paulo Bauru School of Dentistry (SP, Brazil), with BOP < 1, PD < 2 mm, and CAL < 1 mm (*n* = 163).

All subjects signed an informed consent, and research protocols were reviewed by the corresponding institutional ethics committee. It is critically important to emphasize that all recruited subjects were non-smokers (including former smokers), as tobacco use is a major environmental risk factor for periodontitis and is nearly impossible to control for in logistic regression models applied to relatively small sample sizes.

### 2.2. Genotyping

Participants’ saliva was collected in an Oragene OG-500 collection kit (DNAGenoTek, Ottawa, ON, Canada). DNA was extracted from saliva using the DNA Investigator Kit (QIAGEN, Hilden, Germany) in an automated QIAcube (QIAGEN) platform, following the manufacturer’s instructions. DNA purity and concentration were assayed in a NanoDrop 2000 spectrophotometer (ThermoFisher, Waltham, MA, USA). Extracted DNA was immediately used for genotyping. Allelic discrimination for SNP rs4986790 was performed in 384-well plates in 5 μL duplicate reactions using Taqman (Applied Biosystems, Warrington, UK) chemistry, as previously described. Genotyping was blinded to group status. For reaction quality control, a sample of known genotype was included in the plate, and a no-DNA template sample was included as a negative control. Samples that failed to provide a genotype were repeated in additional reactions. Genotyping procedures, from isolation to allelic discrimination, and data analysis were performed either in the Osteoinmunology Laboratory, Bauru Faculty of Dentistry, University of São Paulo, or at the Center for Cranio Facial Research, University of Texas Health Science Center in Houston.

### 2.3. Immune Localization of TLR4 and NFκB in Mice Tissues

As a first step in the functional characterization of the effect of SNP rs4986790, we performed an immune localization of TLR4 and NFκB in mice periodontal tissues. NFκB is the main transcription factor that gets translocated to the nucleus and activated after TLR4 stimulation [15].

Mice maxillae were obtained from adult 8-week-old male C57/Bl6 mice and fixed in PBS-buffered formalin 10% pH 7.4. Tissues were demineralized in EDTA 4% for 4 weeks with weekly changes of EDTA. The demineralized tissues were formalin-fixed paraffin-embedded and sectioned into 6 µm thick sections for standard hematoxylin and eosin staining and immunolocalization of TLR4 and NFκB. The sectioning was performed in the incisor region of the maxillae. Sections were cleared in xylene and rehydrated in gradate alcohol baths. Antigen retrieval was performed in 10 mM sodium citrate buffer (pH 6) at 96 degrees C for 5 min. Sections were then permeabilized with 0.5% Triton X-100 in PBS, blocked with 1% BSA in 10% goat serum, and incubated overnight at 4 degrees C with rat anti-mouse TLR4 (R&D system, # MAB2759-SP, Minneapolis, MN, USA), rabbit anti-human/mouse p65 NFκB (EDM Millipore, #PC138, Darmstadt, Germany), followed by the incubation with secondary antibodies Alexa Fluor^®^555 goat anti-rat (Life Technologies, #A2120, Carlsbad, CA, USA) and Alexa Fluor^®^488 donkey anti-rabbit (Life Technologies, #A21206). DAPI was used for nuclear staining. Images were acquired in a Nikon Eclipse Ni-U upright fluorescence microscope (Nikon, Tokyo, Japan) equipped with a Zyla 5.5 sCMOS camera (Andor Technologies, Belfast, UK).

### 2.4. Transfection of Asp299Gly Variant of TLR4 Receptor

The second step in the functional description of the effect of SNP rs4986790 was to perform transient transfection of the wild type and polymorphic forms of the gene into several cell lines and characterize their response to LPS stimulation. Additionally, we treated the cell cultures with TLR4 pathway inhibitors to unveil the possible mechanism of the differential response of the truncated protein. Each inhibitor was incubated with optimized timing based on preliminary tests.

We obtained the plasmids hu-TLR4299snp-flagpDEST40 (#42647) and hu-TLR4cDNAwtpDEST40 (#42646) from the Addgene plasmid repository [16]. The empty vector backbone pDEST40 was purchased from ThermoFisher Scientific.

Human embryonic kidney cells (293T) were purchased from American Type Culture Collection (Gaithersburg, MD, USA) and cultured in Dulbecco’s modified Eagle’s medium (Gibco, Grand Island, NY, USA), supplemented with 10% fetal bovine serum (Gibco) and 1% penicillin/streptomycin (Gibco) at 37 C 5% CO2 atmosphere.

Human fibrosarcoma cells (HT1080) were purchased from the American Type Culture Collection (Gaithersburg, MD, USA) and cultured in Eagle’s minimal essential medium (Gibco, Grand Island, NY, USA), supplemented with 10% fetal bovine serum (Gibco) and 1% penicillin/streptomycin (Gibco) at 37 C 5% CO2 atmosphere.

Cells were seeded at a density of 15.000 cells/well in a 96-well plate (Corning, Kennebunk, ME, USA) and cultured for 24 h to reach 80% confluence. Cells were transfected with 50 ng of each construct in a 3:1 ratio with FuGENE HD transfection reagent (Promega, Madison, WI, USA) with OptiMEM medium (Life Technologies, Grand Island, NY, USA) and cultured for 48 h.

Cell cultures were stimulated with *E. coli* LPS 8 µg/mL (SigmaAldrich, Burlington, MA, USA) 12 h after the transfection protocol and maintained in culture conditions for an additional 24 h. *E. coli* LPS was chosen due to its wide use as a standard model for Gram-negative bacterial activation in periodontal research, facilitating comparison with the existing literature. Before the LPS stimulation, selected wells were pretreated with TLR4 pathway inhibitors.: IAXO-102 5µM (Adipogen, San Diego, CA, USA), TAK 242 100 nM (Cayman Chemical, Ann Arbor, MI, USA), and SN50 10 µM (Calbiochem, Darmstadt, Germany). Concentration and timing were tested and optimized in previous experiments. All experiments were performed in triplicate and repeated independently three times. The mechanism of action of each inhibitor is depicted in Figure 1.

Supernatant was collected and mixed in a 10:1 ratio in Roche EDTA-free protease inhibitor cocktail (Indianapolis, IN, USA), snap frozen in liquid nitrogen, and stored in −80 degrees Celsius. Lysates were obtained using RIPA buffer (ThermoFisher) following the manufacturer’s directions. Lysates were mixed in a 10:1 ratio in Roche EDTA-free protease inhibitor cocktail (Indianapolis, IN, USA), snap frozen in liquid nitrogen, and stored in −80 degrees Celsius.

Collected supernatants were concentrated 5× using Amicon Ultra-4 3000 Da MWCO centrifugal filter units following the manufacturer’s directions. Lysates were centrifuged at 12000× *g* for 5 min at 4 °C, and supernatants were collected and mixed in a 10:1 ratio in Roche EDTA-free protease inhibitor cocktail. The samples were then titled for IL-6, IL-8, SDF-1 (CXCL12), and TNFα using ELISA kits specific for cell culture supernatant or cell lysate (RayBiotech, Norcross, GA, USA) following the manufacturer’s instructions.

Cytokine concentration data were tested for normal distribution using the Shapiro–Wilk test, and differences between groups and control were assessed by ANOVA post hoc Dunn’s test. Cells transfected with the empty vector were included as a technical control in all experiments. A *p*-value < 0.05 was set as indicative of a significant difference.

Additional cultures were performed in 24-well plates to conduct immunocytofluorescent localization of flag-tag TLR4 to test transfection efficiency and NFκB to evaluate nuclear translocation. In selected experiments, we also localized the native TLR4 protein. Briefly, a sterile cover slip was placed inside the 24-well culture plates, rinsed with PBS 1×, followed by a quick rinse in culture media, and then the cells were seeded at 5 × 10^5^ cells/well and cultured for 24 h. Cells were transfected and/or treated with TLR4 pathway inhibitors, and after 48 h, the cells were washed with cold 1× PBS, then fixed with 4% formaldehyde (freshly made) for 35 min and gently washed 2 times with 1× PBS for 2 min. Then, the cells were incubated with 0.1 M glycine solution for 15 min, and subsequently, they were washed with 1× PBS for 5 min. Later, the cells were incubated with a blocking solution (1% bovine serum albumin diluted in 1× PBS) of 0.1% Triton X-100 for 1 h at RT. The primary antibodies were diluted in the blocking solution at a 1:100 concentration: DYKDDDDK Tag Monoclonal Antibody Alexa Fluor 555 (ThermoFisher #MA1-142-A555, Waltham, MA, USA), rat anti-mouse TLR4 (R&D system, # MAB2759-SP), and/or rabbit anti-human/mouse p65 NFκB (EDM Millipore, #PC138). After incubation, repeated washing steps were performed with 1× PBS (3 times, 10 min each). Then, the cells were incubated with secondary antibody Alexa Fluor^®^488 donkey anti-rabbit (Life Technologies, #A21206), and/or Alexa Fluor^®^555 goat anti-rat (Life Technologies, #A2120) diluted at 1:150 in the blocking solution, incubated in a dark chamber, for 2 h at RT. Then, cells were stained with DAPI (ThermoFisher^®^, #D3571) diluted at 3µM in ddH2O for 10 min. The cover slides were mounted on a conventional histological slide with ProLong Gold Antifade Reagent (Invitrogen, P36930, Waltham, MA, USA). Imaging was performed 48 h after mounting steps, in a Nikon Eclipse Ni-U upright fluorescence microscope (Nikon Instruments) equipped with a Zyla 5.5 sCMOS camera (Andor, Belfast, UK).

## 3. Results

### 3.1. Genotyping

We aimed to demonstrate an association between rs4986790 and a decreased occurrence of inflammatory periodontal destruction. Association analyses were performed in one exploratory population and three independent validation populations.

In the exploratory population (*n* = 570, 186 cases/384 controls), rs4986790 demonstrated a significant protective effect for the phenotype of chronic periodontitis [OR 0.3315, CI 0.14–0.75, *p*-value 0.005] (Table 1).

The three validation populations were tested independently for association with the disease phenotype. The PIT and SAO populations did not demonstrate a significant association between the occurrence of the polymorphic genotype for rs4986790 and a differential risk of suffering chronic periodontitis, while the HOU population demonstrated a significant association between polymorphic genotype and disease phenotype (Table 2).

The association analysis combining exploration and all three validation populations demonstrated a significant protective effect of the polymorphic allele for the phenotype ‘susceptibility for inflammatory alveolar bone resorption’ [OR 0.57, CI 0.38–0.85, *p*-value 0.005] (Table 3).

### 3.2. Immune Localization of TLR4 and NFκB

To support a mechanistic explanation for the observed genetic association, we conducted complementary experiments to confirm the presence and expression of TLR4 in periodontal tissues. We also localized NFκB, a key transcription factor activated downstream of TLR4 signaling and central to the regulation of inflammatory responses. In the incisor region, both NFκB and TLR4 signals appeared evenly distributed in the periodontal ligament. In most cases, NFκB and TLR4 colocalized (Figure 2 and Figure 3).

The immunolocalization data clearly demonstrated that TLR4 is ubiquitously and constitutively expressed in the periodontium, and that NFκB tends to localize within the same cells expressing TLR4. These findings support a functional link between TLR4 and NFκB in periodontal tissues, consistent with the proposed role of TLR4 signaling in modulating local inflammatory responses.

### 3.3. Transfection of Asp299Gly Variant of TLR4 Receptor

To investigate the functional impact of the TLR4 Asp299Gly polymorphism, we transfected different cell lines with plasmids encoding either the wild-type (WT) or the mutant variant of TLR4. This approach allowed us to assess the effect of the mutation on downstream signaling and cytokine responses under controlled experimental conditions.

HEK293T control cells (no transfection) did not exhibit a fluorescent signal for the flag-tag (TLR4), as expected. LPS-treated control cells exhibited a nuclear translocation of NFκB, while non-LPS-treated cells demonstrated a faint NFκB signal, with cytoplasmic accumulation (Figure 4).

Wild-type (WT) and mutant (MT) transfected HEK293T cells ubiquitously expressed the flag-tagged TLR4 protein, demonstrating a high efficiency of the transfection protocol. It was impossible to detect transfected cells not evidencing flag-tag signals. WT and MT transfected cells significantly increased the expression of TLR4 after LPS stimulation and demonstrated a more evenly distributed signal that is compatible with membrane expression. Further, after LPS stimulation, it was possible to observe an increased NFκB signal and evidence of nuclear accumulation. From a qualitative perspective, there were no evident differences between WT and MT transfected cells (Figure 4).

IL-8 titration of transfected HEK293T cell culture supernatant demonstrated significantly increased secretion of IL-8 for hu-TLR4299snp-flagpDEST40 transfected cells. This data seem to support increased LPS sensitivity of the cells carrying the polymorphic version of the TLR4 receptor (Figure 5).

IL-6, SDF-1 (CXCL12) and TNFα were undetectable in HEK293T supernatants. Cell lysates were not tested in this cell type.

Consistent with our results from the HEK293T cell line, HT1080 control cells (no transfection) did not exhibit a fluorescent signal for the flag-tag (TLR4). LPS-treated control cells exhibited a nuclear translocation of NFκB, while non-LPS-treated cells demonstrated a faint NFκB signal, with cytoplasmic accumulation (Figure 6).

Wild-type (WT) and mutant (MT) transfected HT1080 cells ubiquitously expressed the flag-tagged TLR4 protein, demonstrating a high efficiency of the transfection protocol. It was impossible to detect transfected cells not exhibiting the flag-tag signal. WT and MT transfected cells significantly increased the expression of TLR4 after LPS stimulation and demonstrated a more evenly distributed signal, which is compatible with membrane expression. Transfected but unstimulated cells demonstrated a very faint NFκB signal, apparently exclusively located in the cytoplasm. After LPS stimulation, it was possible to observe an increased NFκB signal and evidence of perinuclear accumulation and nuclear translocation. From a qualitative perspective, MT LPS-treated cells exhibited increased colocalization of TLR4 and NFκB in a perinuclear location (Figure 6). 

IL-6 titration of transfected HT1080 cell culture supernatant demonstrated significantly increased secretion of IL-6 for hu-TLR4299snp-flagpDEST40 and hu-TLR4cDNAwtpDEST40 transfected cells. For each transfection vectors, the cells increased their IL-6 secretion between 2- and 4-fold. TNFα titration in cell culture lysates did not exhibit any significant change in the tested experimental condition relative to the control (Figure 7).

To further elucidate the mechanism underlying the observed functional differences between the wild-type and mutant TLR4, we treated the transfected cells with three different TLR4 pathway inhibitors. This strategy aimed to dissect the contribution of specific components of the TLR4 signaling pathway to the altered cellular responses associated with the polymorphism.

The inhibitory capacity of IAXO-102 5µM, TAK 242 100 nM, and SN50 10 µM was evidenced by the immune localization of NFκB in HT1080 cells. Untreated control cells demonstrated a faint fluorescent signal of TLR4 and an almost undetectable signal of NFκB. After the stimulation with LPS 8 µg/mL, there was an evident increase in the expression of TLR4, with the fluorescent signal evenly distributed in the cell. NFκB expression was apparent in all cell compartments, with clear evidence of nuclear translocation (Figure 8). The cell cultures pretreated with the TLR4 inhibitor TAK-242 demonstrated decreased TLR4 and NFκB expression and cytoplasmic accumulation of NFκB, with no evidence of NFκB nuclear translocation. IAXO-102 and SN50 treatments produced similar results, with the strongest inhibitory effect produced by SN50 (Figure 8).

IL-6 titration in cell culture supernatant dropped significantly in LPS-treated cells, with the IAXO-102 pretreatment in the empty vector backbone pDEST40 transfection group, while the other two inhibitors seemed to produce no effect on the IL-6 secretion (Figure 9). In the hu-TLR4cDNAwtpDEST40 (WT) transfected group, IL-6 increased evenly by ±4-fold compared with the empty vector group in all four experimental groups, with no evidence of inhibitory effect of any of the TLR4 pathway inhibitor compounds (Figure 9). Conversely, in the hu-TLR4299snp-flagpDEST40 (MT) transfected group, there was a significant inhibitory effect of IAXO-102, like the effect observed in the control empty vector transfection group. As was the case in the WT transfected group, the levels of secretion of IL-6 increased by ±4-fold with respect to the control empty vector transfection group. Apparently, the MT version of the TLR4 receptor is more sensitive to the inhibitory effect of the CD14 inhibitor IAX0-102 than the WT form of the receptor.

TNFα titration in cell lysates demonstrated no variation in any of the experimental conditions tested. Expression levels were maintained in the empty vector backbone pDEST40, hu-TLR4cDNAwtpDEST40, and hu-TLR4299snp-flagpDEST40 transfected cells. Similarly, the inhibitors seemed to exert no effect on the expressed levels of TNFα (Figure 10). This result was unexpected since published evidence suggested that TNFα is differentially expressed by cells carrying the rs4986790 SNP [17]. This can be explained by the fact that we assayed cell lysates instead of cell culture supernatants or by an idiosyncratic peculiarity of the cell type used in the experiment.

## 4. Discussion

Globally, our results point to a significant contribution of the polymorphism rs4986790 in the TLR4 gene in the pathogenesis of apical and chronic periodontitis. Our preliminary data from the Brazilian cohort indicating a protective effect of the polymorphic form of the gene was corroborated in an independent cohort of patients recruited from a distinct population.

A previous meta-analysis including *n* = 2879 (1345 cases) suggested that there was no discernible effect of rs4986790 polymorphism in periodontitis pathogenesis. Nevertheless, it must be considered that this meta-analysis included populations from various ethnic backgrounds, including European Caucasians, Asians, Africans, and mixed Latin-American population [18]. The discrepancy with our current results could be the result of a differential effect of the genetic polymorphism in populations with different backgrounds and by the interaction of other polymorphisms inherited in linkage disequilibrium in a differential fashion between the dissimilar populations [18,19].

We believe that our association results are trustworthy since the disease phenotype is extremely well characterized by experienced clinicians and because we obtained replication of our association in a validation population. Although sex was recorded for all participants, the study was not powered to detect sex-specific effects, and potential differences related to sex could not be assessed. This limitation should be addressed in future studies with adequate stratification.

The functional experiments unequivocally demonstrated that the polymorphism rs4986790 has functional effects in the sensitivity of the TLR4 receptor to LPS-stimulation. HEK293T cells transfected with the polymorphic version of the TLR4 receptor demonstrated increased secretion of IL-8 under LPS stimulation. This result is in accordance with reports from peripheral blood mononuclear cells obtained from heterozygous individuals for rs4986790, which demonstrated increased cytokine production under LPS stimulation [18]. Similarly, the polymorphic form of the TLR4 receptor seemed to have increased sensitivity to CD14 inhibition than the wild-type form. Published evidence suggests that the polymorphic form of the receptor has structural changes near the region where it interacts with MD-2 during the transference of LPS from CD14 [20]. This could be the reason explaining the differential susceptibility of the truncated receptor to the CD14 inhibitor.

It is important to emphasize that the immunolocalization images of TLR4 and NFκB are presented as supportive observations rather than central evidence for our conclusions. Although these images can help support our conclusions, the primary findings of this study are based on the genetic association analysis. Despite their limited resolution, the images are consistent with the proposed involvement of TLR4 signaling in periodontal inflammation. Further studies are needed to elucidate the functional implications of the TLR4 polymorphism in this context.

## 5. Conclusions

Our study provides strong evidence that the TLR4 Asp299Gly single-nucleotide polymorphism (rs4986790) is associated with reduced susceptibility to periodontal inflammatory destruction. The protective effect of the polymorphic allele was consistently observed across multiple independent populations, reinforcing the robustness of this genetic association. Functional in vitro analyses revealed that cells expressing the polymorphic variant exhibit increased IL-8 secretion in response to LPS stimulation and heightened sensitivity to CD14 inhibition. These findings indicate that the structural alteration introduced by the SNP modulates TLR4 signaling, potentially resulting in a more controlled immune response within the periodontal environment. Collectively, our results support a functional role for rs4986790 in shaping host–microbial interactions and influencing individual susceptibility to periodontitis. Further research is warranted to assess the clinical relevance of this variant and its potential implications for precision periodontics.

## Figures and Tables

**Figure 1 biology-14-00894-f001:**
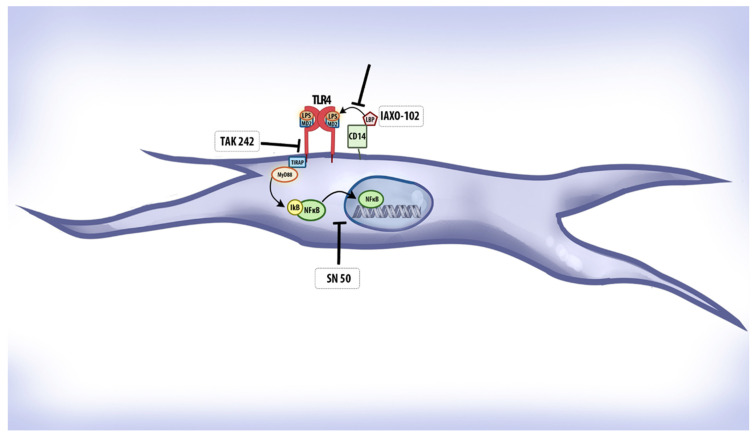
The TLR4 functional receptor complex is composed of TLR4 itself, the co-receptor CD14, and the accessory protein MD-2. Upon recognition of lipopolysaccharide (LPS) from Gram-negative bacteria, CD14 facilitates the transfer of LPS to the TLR4/MD-2 complex, promoting TLR4 dimerization and activation. This leads to the recruitment of intracellular adaptor proteins, primarily MyD88, initiating a signaling cascade that culminates in the nuclear translocation of NFκB and the transcription of proinflammatory cytokines. In this study, three inhibitors were used to target distinct steps of this pathway. IAXO-102 is a small-molecule antagonist that binds to CD14, competitively inhibiting its interaction with LPS and, thereby, blocking the delivery of LPS to the TLR4/MD-2 complex. TAK-242 (resatorvid) binds selectively to the intracellular Toll/IL-1 receptor (TIR) domain of TLR4, preventing the recruitment of adaptor proteins such as MyD88 and TRIF, thereby inhibiting downstream signal transduction. SN50 is a synthetic cell-permeable peptide that mimics the nuclear localization sequence (NLS) of NFκB; it acts by competitively inhibiting the nuclear import of the NFκB complex, effectively blocking the transcription of target genes. This diagram illustrates the specific sites of action of these inhibitors along the TLR4 signaling pathway.

**Figure 2 biology-14-00894-f002:**
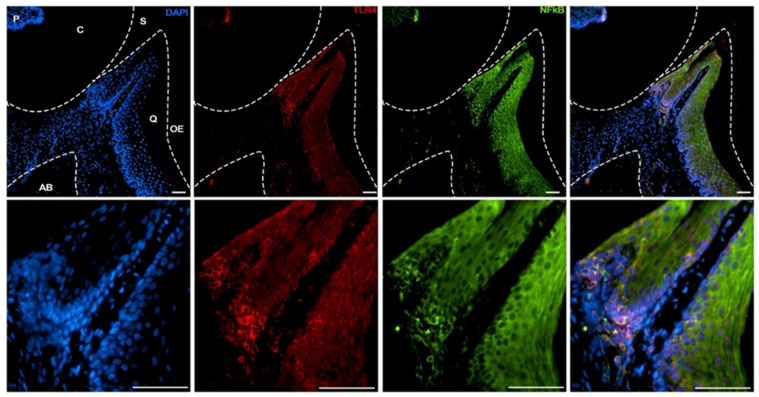
Immune localization of TLR4 and NFκB in the periodontium of the incisor region. *p* = pulp; C = crown; S = sulcus; Q= queratinized epithelium; OE = oral epithelium; AB = alveolar bone. Scale bar= 50 µm. The yellow signal indicates colocalization.

**Figure 3 biology-14-00894-f003:**
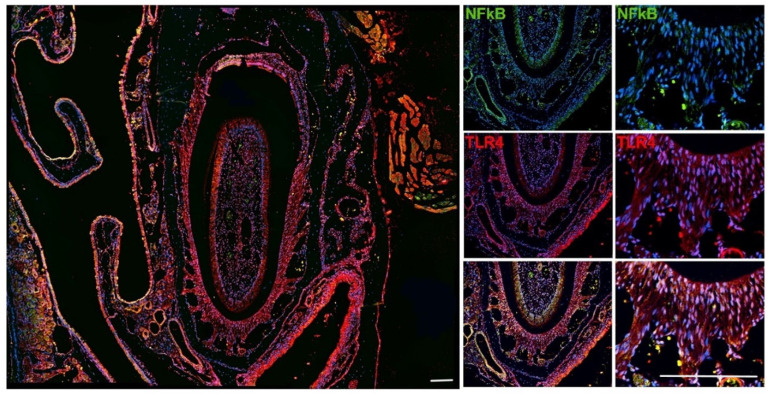
Immune localization of TLR4 and NFκB in the periodontium of the incisor region. Scale bar = 100 µm.

**Figure 4 biology-14-00894-f004:**
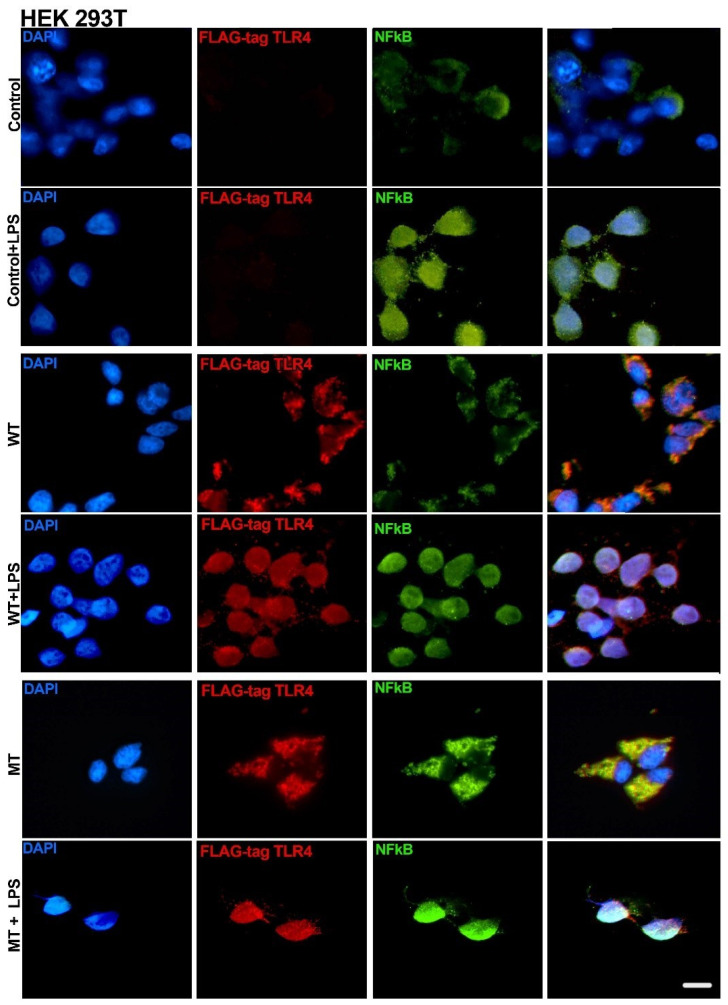
Immune cytofluorescence of the HEK293T cell line. Control, no transfection; WT transfected with hu-TLR4cDNAwtpDEST40; MT transfected with hu-TLR4299snp-flagpDEST40. Cells were treated with *E. coli* LPS (8 µg/mL) for 24 h, starting 12 h after transfection. FLAG-tag (DYKDDDDK) protein tag included in the TLR4-transfected sequence. Scale bar = 20 µm.

**Figure 5 biology-14-00894-f005:**
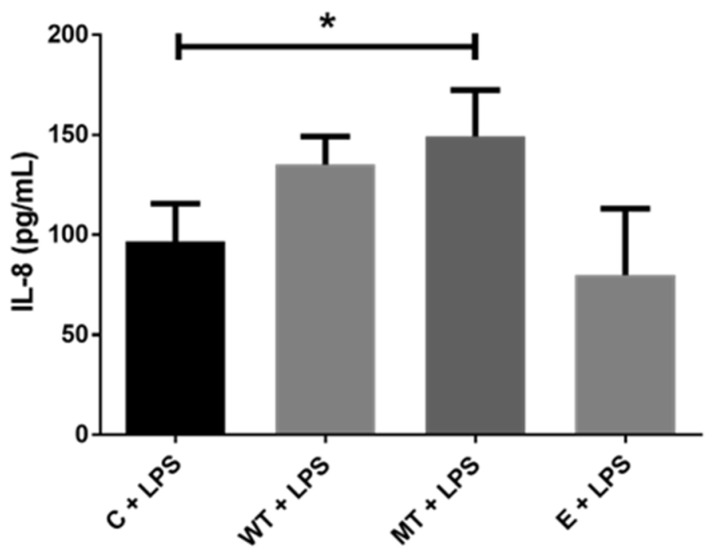
IL-8 supernatant titration in HEK293T transfected cells. Control (C), no transfection; WT transfected with hu-TLR4cDNAwtpDEST40; MT transfected with hu-TLR4299snp-flagpDEST40; empty (E), transfected with the empty backbone plasmid vector pDEST40. LPS treatment 8 µg/mL. * *p* < 0.05 (ANOVA post hoc Dunn’s test). The bar represents the mean, and the error bar represents the SD.

**Figure 6 biology-14-00894-f006:**
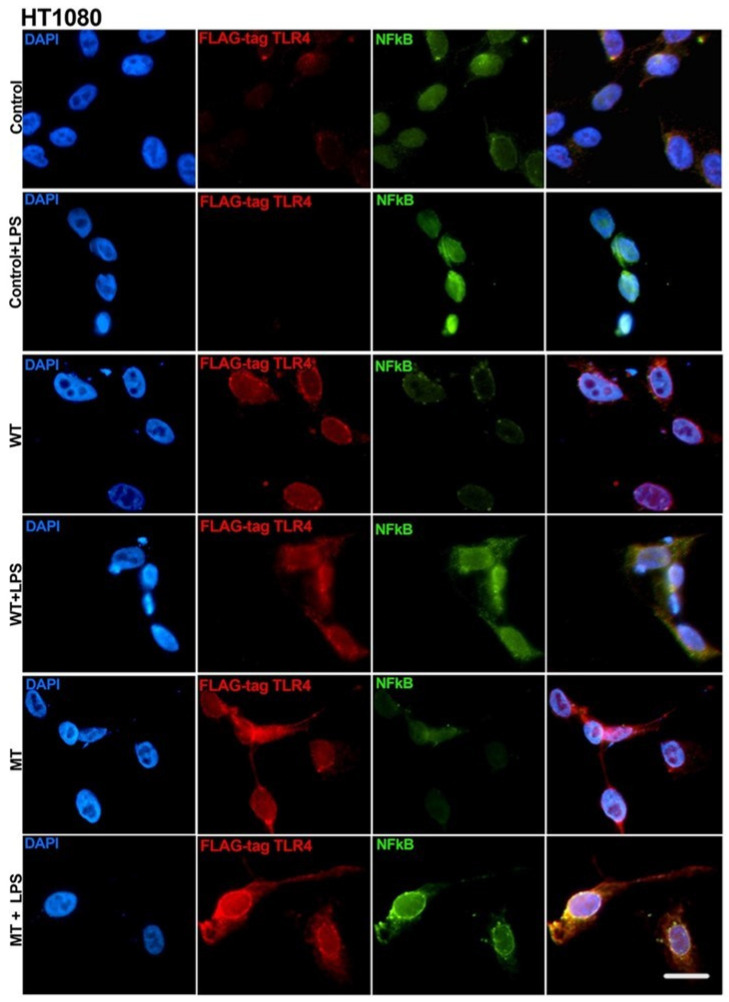
Immune cytofluorescence of the HT1080 cell line. Control, no transfection; WT transfected with hu-TLR4cDNAwtpDEST40; MT transfected with hu-TLR4299snp-flagpDEST40. Cells were treated with *E. coli* LPS (8 µg/mL) for 24 h, starting 12 h after transfection. FLAG-tag (DYKDDDDK) protein tag included in the TLR4-transfected sequence. Scale bar = 20 µm.

**Figure 7 biology-14-00894-f007:**
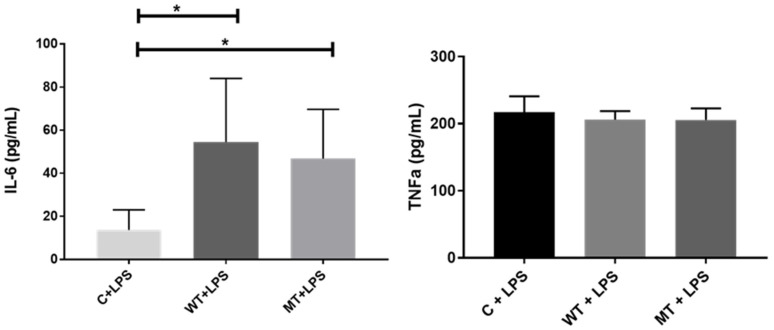
IL-6 supernatant titration in HT1080 transfected cells and TNFα cell culture lysate titration in HT1080 transfected cells. Control (C), no transfection; WT transfected with hu-TLR4cDNAwtpDEST40; MT transfected with hu-TLR4299snp-flagpDEST40. LPS treatment 8 µg/mL. * *p* < 0.05 (ANOVA post hoc Dunn’s test). The bar represents the mean, and the error bar represents the SD.

**Figure 8 biology-14-00894-f008:**
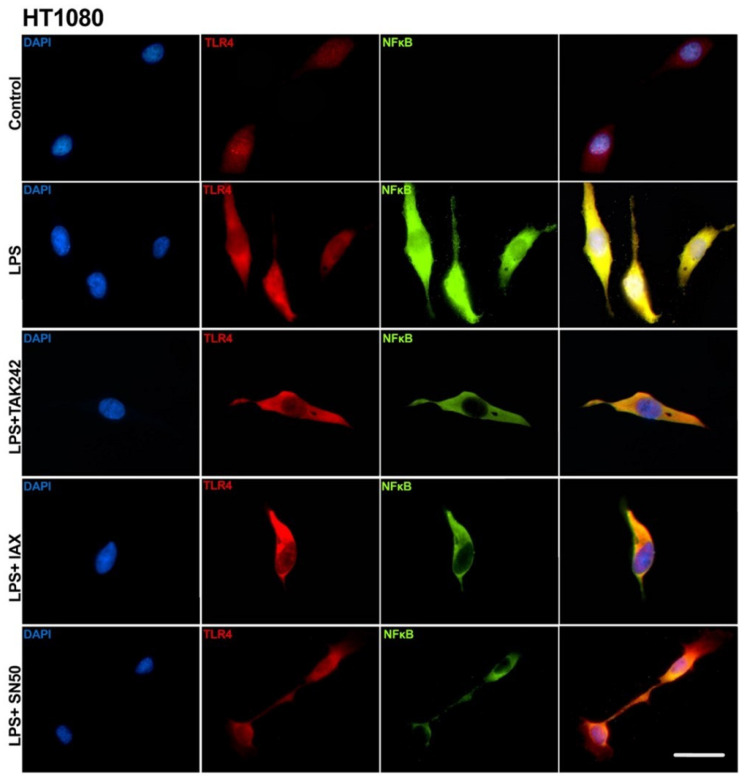
Immune cytofluorescence of TLR4 and NFκB in the HT1080 cell line. Control, no LPS-treatment. Cells were treated with *E. coli* LPS (8 µg/mL) for 24 h. LPS + TAK242= *E. coli* LPS pretreated with TAK 242 100 nM. LPS + IAX = *E. coli* LPS pretreated with IAXO-102 5µM. LPS + SN50 = *E. coli* LPS pretreated with SN50 10 µM. Scale bar = 20 µm.

**Figure 9 biology-14-00894-f009:**
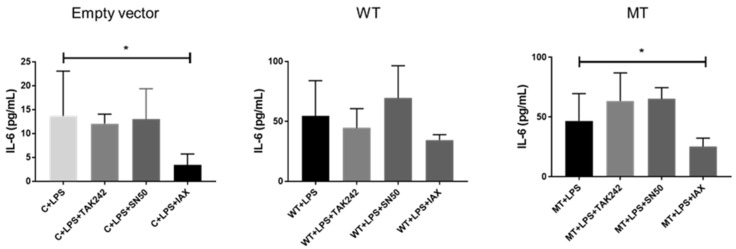
IL-6 supernatant titration in HT1080 transfected cells. Empty vector transfected with empty vector backbone pDEST40; WT transfected with hu-TLR4cDNAwtpDEST40; MT transfected with hu-TLR4299snp-flagpDEST40. LPS stimulated with 8 µg/mL *E. coli* LPS. LPS + TAK242 = 8 µg/mL *E. coli* LPS pretreated with TAK 242 100 nM. LPS + IAX = 8 µg/mL *E. coli* LPS pretreated with IAXO-102 5 µM. LPS + SN50 = 8 µg/mL *E. coli* LPS pretreated with SN50 10 µM. * *p* < 0.05 (ANOVA post hoc Dunn’s test). The bar represents the mean, and the error bar represents the SD.

**Figure 10 biology-14-00894-f010:**
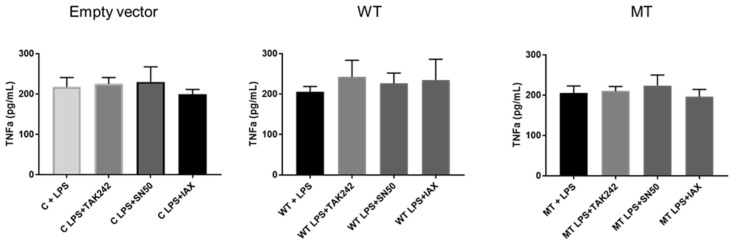
TNFα cell lysate titration in HT1080 transfected cells. Empty vector transfected with empty vector backbone pDEST40; WT transfected with hu-TLR4cDNAwtpDEST40; MT transfected with hu-TLR4299snp-flagpDEST40. LPS stimulated with 8 µg/mL *E. coli* LPS. LPS + TAK242 = 8 µg/mL *E. coli* LPS pretreated with TAK 242 100 nM. LPS + IAX = 8 µg/mL *E. coli* LPS pretreated with IAXO-102 5µM. LPS + SN50 = 8 µg/mL *E. coli* LPS pretreated with SN50 10 µM. The bar represents the mean, and the error bar represents the SD.

**Table 1 biology-14-00894-t001:** Association analysis of rs4986790 and chronic periodontitis in an exploratory population (*n* = 570, 186 cases/384 controls). OR = odds ratio; SE = standard error; 95% CI = 95% confidence interval.

Frequency Cases	Frequency Controls	Chi Square	*p*-Value	OR	SE	95% CI
0.01882	0.05469	7.839	0.005113	0.3315	0.4133	0.14–0.75

**Table 2 biology-14-00894-t002:** Association analysis of rs4986790 and chronic periodontitis/apical periodontitis in exploratory and validation populations. OR = odds ratio; SE = standard error; 95% CI = 95% confidence interval.

	Frequency Cases	Frequency Controls	Chi Square	*p*-Value	OR	SE	95% CI
Exploration (RBP)	0.01882	0.05469	7.839	0.0051	0.3315	0.4133	0.14–0.75
Validation 1 (PIT)	0.03226	0.05278	0.858	0.3543	0.5982	0.5603	0.19–1.79
Validation 2 (HOU)	0.03741	0.1208	13.27	0.0002	0.2828	0.3656	0.13–0.57
Validation 3 (SAO)	0.05963	0.02813	3.282	0.07	2.191	0.4429	0.91–5.22

**Table 3 biology-14-00894-t003:** Association analysis of rs4986790 and ‘susceptibility for inflammatory alveolar bone resorption’ in pooled population (*n* = 1410; 528 cases and 882 controls). OR = odds ratio; SE = standard error; 95% CI = 95% confidence interval.

Frequency Cases	Frequency Controls	Chi Square	*p*-Value	OR	SE	95% CI
0.03472	0.05865	7.65	0.005678	0.5774	0.2008	0.38–0.85

## Data Availability

The data presented in this study are available on request from the corresponding author. The data are not publicly available due to institutional policy.

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
