# Peer review of "TLR4 Asp299Gly SNP (rs4986790) Protects from Periodontal Inflammatory Destruction by Altering TLR4 Susceptibility to LPS Stimulation"

_biology, 2025, doi:10.3390/biology14070894_

Round 1
Reviewer 1 Report
Comments and Suggestions for Authors
In this study, Cavalla et al. investigated the role of TLR4 Asp299Gly SNP rs4986790 in the association with periodontitis. Although there was variation across the analyzed populations, the authors found that the SNP has a potential association with chronic periodontitis. They also tried to show the localization of TLR4 and NFkB in the periodontal ligament to support the importance of TLR4 signaling in preventing periodontitis. Additionally, they investigated TLR4 signaling in response to LPS from WT and mutant TLR4-expressing cells by showing the cellular localization of TLR4 and NFkB and analyzing inflammatory cytokines.
Overall, their findings look interesting. However, the manuscript lacks sufficient experimental evidence to support the key claims. A more comprehensive analysis would be necessary for consideration in Biology.
(Comments)
It would be beneficial to have editorial support to correct typos and grammar and improve overall clarity.
Please provide details about the statistical analysis from each figure, including the type of statistical test used, the type of error bars (e.g., SD or SEM) and whether the values represent the mean or median, either in the Materials and Methods section or in the figure legends.
Results section: It would be beneficial to state the author’s hypothesis or rationale for conducting each specific analysis or experiment at the beginning of each results section. As it stands, the explanation is somewhat difficult to follow.
Lines 252-253: Please consider revising the figure citation. The legend for Figure 2 indicates that the image is from the molar region, whereas the text describes the data from the incisor region.
Figure 2: It would be helpful to point out where TLR4 and NFkB are co-localized in the image. It’s not easy to find the co-localization of two proteins.
Please add a scale bar to every microscopic image. (Figure 3, Figure 4, Figure 6, Figure 8)
In the figures showing experimental results treated with LPS, please specify the duration of LPS treatment for the analysis.
The overall quality of the images showing the localization of TLR4 and NFkB is not great. It’s not sufficient to support the author’s statement as they are.
Reviewer 2 Report
Comments and Suggestions for Authors
What is the validation sample mentioned in line 73? Are they additional samples from different places?
Did you consider a sex based difference in the study?
In the figure 1 diagram, it’s not very clear where the IAX0-102 acts, although the figure caption explains it. Could you move around where the IAX0-102 is placed in the figure to bring more clarity to the readers?
Is the incubation time of the different TLR4 pathway inhibitors the same or was it optimized separately for different inhibitors as mentioned in Line 210.
What is the magnification in the second panel of Figure 02?
Reviewer 3 Report
Comments and Suggestions for Authors
1. Abstract: If there is more space to write in the abstract, please state in the introduction, before the objective, the relevance or reason for studying the TLR4 Asp299Gly SNP (rs4986790) in periodontitis. The methodology does not describe the use of LPS, TLR4, or the samples used.
2. Write the objective and hypothesis of the study in the introduction.
3. Methodology: Specify the area from which the samples were obtained and how they were obtained. The methodology is difficult to follow, so please include a diagram explaining where the samples were taken, which experiments were performed, and the number of patients.
4. Discussion: Discuss the choice of E. coli LPS. Compare results with other studies based on the results shown in the respective sorted section.
Round 2
Reviewer 1 Report
Comments and Suggestions for Authors
The authors have adequately addressed the comments. It seems the manuscript is ready for publication.